# Optimization of Glutathione Adhesion Process to Modified Graphene Surfaces

**DOI:** 10.3390/nano11030756

**Published:** 2021-03-17

**Authors:** Witold Jakubowski, Radomir Atraszkiewicz, Dorota Nowak, Damian Batory, Witold Szymański, Anna Sobczyk-Guzenda, Łukasz Kaczmarek, Piotr Kula, Marian Cłapa, Tomasz Warga, Małgorzata Czerniak-Reczulska

**Affiliations:** 1Division of Biophysics, Institute of Materials Science and Engineering, Lodz University of Technology, 1/15 Stefanowskiego St., 90-924 Lodz, Poland; 2Division of Surface Engineering and Heat Treatment, Institute of Materials Science and Engineering, Lodz University of Technology, 1/15 Stefanowskiego St., 90-924 Lodz, Poland; radomir.atraszkiewicz@p.lodz.pl (R.A.); piotr.kula@p.lodz.pl (P.K.); 3Division of Biomedical Engineering and Functional Materials, Institute of Materials Science and Engineering, Lodz University of Technology, 1/15 Stefanowskiego St., 90-924 Lodz, Poland; dorota.nowak@tzmo-global.com (D.N.); marian.clapa@p.lodz.pl (M.C.); malgorzata.czerniak-reczulska@p.lodz.pl (M.C.-R.); 4Department of Vehicles and Fundamentals of Machine Design, Lodz University of Technology, 1/15 Stefanowskiego St., 90-924 Lodz, Poland; damian.batory@p.lodz.pl; 5Division of Nanomaterials Engineering, Institute of Materials Science and Engineering, Lodz University of Technology, 1/15 Stefanowskiego St., 90-924 Lodz, Poland; witold.szymanski@p.lodz.pl; 6Division of Coating, Polymer and Non-Metal Engineering, Institute of Materials Science and Engineering, Lodz University of Technology, 1/15 Stefanowskiego St., 90-924 Lodz, Poland; anna.sobczyk-guzenda@p.lodz.pl; 7Division of Advanced Materials and Composite, Institute of Materials Science and Engineering, Lodz University of Technology, 1/15 Stefanowskiego St., 90-924 Lodz, Poland; Lukasz.Kaczmarek@p.lodz.pl (Ł.K.); tomasz.warga@p.lodz.pl (T.W.)

**Keywords:** graphene, biofunctionalization, glutathione, biomedical applications

## Abstract

The presented work shows the results of the functionalization of the graphene surface obtained by the growth on the liquid bimetallic matrices method. We used glutathione (GSH) as a peptide model, which allowed us to optimize the procedure to obtain high process efficiency. To establish the amount of GSH attached to the graphene surface, the Folina-Ciocalteu method was used, which allows the assessment of the concentration of colored reaction products with peptide bonds without the disadvantages of most methods based on direct colored reaction of peptide bonds. Samples surface morphology, quality of graphene and chemical structure in the subsequent stages of surface modification were tested—for this purpose Raman spectroscopy, scanning electron microscopy (SEM), and Fourier-transform infrared spectroscopy (FTIR) were used.

## 1. Introduction

For the last decade, various nanomaterials have been used during the process of designing biosensors for increasing their sensitivity and selectiveness of substrates. The healthcare and medical sciences market has increasingly high requirements towards biosensors—on one hand expected sensitivity on the level of nanomolar concentration, on the other, the low cost of the biosensor and reading equipment.

High hopes for reaching the next level of developing biosensors were raised by the possibility of using graphene and its unique physicochemical properties—high singularity surface, electrical conductivity, and optical transparency [1,2,3]. It must be remembered that possible applications are limited not only to pure graphene; graphene derivatives such as graphene oxide and reduced graphene oxide are also included in the area of interest of biomedical use. Furthermore, graphene derivatives can be functionalized much easier by adding metal ions, molecules of metal oxides, as well as organic and inorganic polymers [4,5,6]. All these adjustments greatly expand the range of modifications of generated graphene nanocomposites, both as 3D and 2D materials, simultaneously opening vast possibilities of detection techniques. In the case of applying functionalized graphene materials, methods based on light detection can be used, in particular spectrophotometric, fluorescent, and electrochemical ones [7,8]. More importantly, these methods can be used concurrently, in addition to supplementary methods or methods allowing simultaneous acquisition of results from separate substrates using the same platform.

A key fact to note is that graphene is characterized by very high biocompatibility, which allows for the design of sensors that can remain in direct contact with tissues and body fluids [9,10]. This, in turn, greatly expands the potential application possibilities by those related to sensitive blood oxygenation or glucose level detectors. Currently a lot of science groups are leading research on using graphene in biosensors with the most important utilizations as glucose, hydrogen peroxide, markers of cancerogenic processes and pathogenic bacteria detection [11,12,13,14]. Numerous strategies of graphene application indicate another aspect as well, namely that different ways of obtaining this carbon material lead to a graphene with subtle differences in properties, despite the high similarity of the analyzed structures.

The most popular method of manufacturing large area graphene is the CVD method where during graphene synthesis processes various metallic growth substrates are used, including thin copper foils [15]. However, the main disadvantage of CVD graphene production is the overlapping phenomenon, which significantly reduces its mechanical strength, while increasing the share of π-π bonds between the graphene layers This effect is mostly due to the inability to achieve an atomically flat substrate structure. Manufacturing of metallurgical graphene (HSMG) is the method developed at Lodz University of Technology (Lodz, Poland), based on the growth of graphene from the liquid phase on bimetallic substrates with no overlapping effect [16,17,18,19]. Another method is the epitaxial growth of low-defect, high-purity graphene on the SiC single-crystal substrate [20]. Separate groups of graphene structures are powders or graphene flakes. The proper nomenclature seems to be n-layered graphene oxide or reduced graphene oxide flakes, manufactured by chemical or chemical ultrasound assisted exfoliation [16,21,22,23,24].

Another key aspect that is important in terms of the possibility of functionalization is the type of activation of the graphene surface. Different methods affect the level of observed carbon ring structure defects, which constitute active centers enabling surface functionalization. In this case, both chemical methods based on the use of strong inorganic acids and methods based on thermal or electrochemical effects are used for activation. All such distinctions of technologies have an impact on surface functionalization, but not only in relation to the possibility of binding individual types of signal molecules, but also on the efficiency of these processes and the durability of such connections.

The presented work shows the results of the functionalization of the graphene surface obtained by the method of growth on liquid bimetallic matrices. We used glutathione as a peptide model, which allowed us to optimize the procedure to obtain high process efficiency. Furthermore, we aimed to determine the impact of storage conditions of the modified surfaces on the durability of the obtained functionalization. To establish the level of glutathione (GSH) attached to the graphene surface, the Folina-Ciocalteu method was used, which allows the assessment of the concentration of colored reaction products with peptide bonds without the disadvantages of most methods based on direct colored reaction of peptide bonds. We showed a different nature of the correlation between the method of surface preparation and the level of bonded glutathione in the case of using graphene obtained with the use of the delamination technique.

## 2. Materials and Methods

### 2.1. Production of Graphene

Copper-nickel metallic composite [17] was used as a substrate for the production of metallurgical graphene. It was heated to the temperature of 1200–1250 °C in argon protective atmosphere at a constant pressure of 100 kPa. Nucleation and growth steps took place in the liquid phase. The mixture of acetylene, ethylene, and hydrogen in the ratio of 2:2:1 was used as a carburizing medium in the temperature range of 1200–1250 °C. Afterwards, the plate was cooled to the temperature of 1050 °C at a cooling speed of 0.5 °C/min, also in argon atmosphere at a constant pressure of 100 kPa [25]. Details of the HSMG graphene manufacturing process can be found in our previous works and patents [16,17,18,19,25,26]. 

### 2.2. Transfer of Graphene to the Substrate

Graphene was transferred from the bimetallic substrate (Cu and Ni) to the target substrate using thin film of polymethyl methacrylate (PMMA). Graphene was released from metallic substrate by two methods: chemical etching and hydrogen delamination. The first method consisted of covering graphene with a thin polymer coating, evaporating the solvent, and then chemically etching the metallic growth substrate. Chemical etching took place in ferric chloride (FeCl_3_). In the next stage of transfer, the foil with graphene was transferred to Si or SiO_2_ substrate. The foil on the target substrate was positioned on the layer of deionized water. Then the set of substrate-graphene and PMMA foil was heated up to approximately 100 °C. 

The second method was hydrogen delamination, which involved the separation of graphene from the growth substrate in an electrochemical process. The growth substrate was used as a cathode in electrochemical process but was previously coated with PMMA polymer. The hydrogen produced at the cathode during electrolysis separated the graphene together with the PMMA foil from the metal surface.

This method has a great advantage, namely high purity of the obtained graphene compared to chemical etching.

### 2.3. Graphene Hydrogenation Process

The graphene transferred on the silicon by the chemical etching was hydrogenated in a plasma-chemical reactor. According to research, exposing graphene to the action of hydrogen plasma influences the electrochemical and physicochemical properties [27,28,29]. Russo and Passmore [30] prove that hydrogen-functionalization of graphene results in a higher adsorption of proteins on the surface of the carbon material, making it a stable substrate for biomolecule detection.

In this paper the plasma source used during the hydrogenation was the glow discharge induced between the two electrodes. During the hydrogenation, the graphene samples were placed on a sample holder installed in the reaction chamber. One of the graphene samples—a “control sample”, was modified with gold electrodes that allowed the resistivity of the graphene to be measured during the hydrogenation process. The resistance measurement signal was led outside the chamber. The hydrogenation process was carried out at a hydrogen pressure ranging from 1 to 2 Pa. The discharge voltage ranged from 500 to 600 V. The discharge current was maintained in the range of 100–400 µA.

The graphene obtained after the hydrogen delamination was not subjected to additional hydrogenation as the hydrogen on the graphene surface was provided by the delamination process itself.

All the samples were then subjected to chemical activation with the use of concentrated acids. 

### 2.4. Chemical Activation of Graphene

Chemical activation of the graphene surface was based on the use of strong inorganic acids to create “bridges” between the graphene and the amino acids. The activation process was conducted with use of sulfuric acid (75% and 50%) (ChemPur, Piekary Slaskie, Poland, CAS Nr 7664-93-9) and nitric acid (50%) (ChemPur, CAS Nr 7697-37-2). An equal volume of concentrated acid or mixture (100 µL) was applied to the surface of the graphene, and then after 60 s the surface was washed with deionized water and dried [31].

### 2.5. Surface Characteristics

Surface morphology, quality of graphene and chemical structure in the subsequent stages of surface modification were tested. For this purpose, the following research techniques were used: Raman spectroscopy, SEM, and FTIR spectroscopy.

#### 2.5.1. Raman Spectroscopy

Analysis of the graphene samples was carried out using an inVia Raman spectroscope (Renishaw, New Mills, UK). Raman spectra were collected using an argon laser with a wavelength λ = 532 nm, the exposure time was 10 s, and the signal was averaged from three repeated expositions per spot. The laser output power was 29.3 mW but during the raman spectra acquisition only 10% of the output power was used. Raman scattering was observed for the 1100–3100 cm^−1^ wavenumber range. The obtained raman spectra were processed using the PeakFit v4.12 software (Systat Software Inc., London, UK). Gauss–Lorentz curves were used for spectra deconvolution. Maximum intensities obtained from the deconvolution were taken for the calculation of characteristic peak ratios of graphene which were used in further analysis.

#### 2.5.2. Scanning Electron Microscopy (SEM)

The structure and morphology of the graphene samples were investigated with use of scanning electron microscope JSM-6610LV (JEOL, Tokyo, Japan) integrated with EDS X-MAX 80 (Oxford Instruments, Oxfordshire, UK) analyzer. The observations were made in high and low vacuum mode using secondary electrons contrast.

#### 2.5.3. Fourier Transform Infrared Spectrometry (FTIR)

Infrared absorption of graphene samples in the spectral range 4000 to 400 cm^−1^ was analyzed with Nicolet model iS50 Fourier transform IR spectrometer (Thermo Scientific company, Waltham, MA, USA). Spectra were recorded with the resolution of 2 cm^−1^ using a high sensitivity DTGS (deuterated triglycine sulfate) detector. The measurements were performed in a reflection mode with an application of a Sequelle DRIFT (Diffuse Reflectance) accessory working at an angle of incidence equal to 10 degrees. In each case, data from 128 scans had been collected to construct a single spectrum.

### 2.6. Glutathione Attachment Procedure

The glutathione attachment procedure was developed by modifying the technique described by L.Mu et al. [32]. The main difference is the replacement of cysteine by a glutathione molecule.

Graphene samples were immersed in a 1ml solution of 1-ethyl-3-(3-dimethylaminopropyl)-carbodiimide (0.2 M) (Sigma Aldrich, Munich, Germany, CAS Nr 25952-53-8) and N-hydroxysulfosuccinimide (0.5 M) (Sigma Aldrich, CAS Nr 106627-54-7) for 3 h (EDAC-NHS mixture). The plates were then moved to solutions containing 40 mM of glutathione (Sigma Aldrich, CAS Nr 70-18-8) and left at room temperature for 24 or 48 h. After this time, the plates were washed three times in 50 mL ddH_2_O (to remove unbound glutathione).

### 2.7. Biochemical Method for Assessing the Level of Glutathione Functionalization

The level of attached glutathione was evaluated using an enzymatic method based on the glutathione reductase reaction [33]. The method has been modified in order to use it in assessing the immobilization of glutathione to graphene. Graphene flakes were immersed in 1000 µL of 100 mM potassium phosphate buffer (ChemPur, CAS Nr 7758-11-4 and CAS Nr 7778-77-0), pH 7.0 containing 1 mM EDTA (Sigma Aldrich, CAS Nr 6381-92-6), followed by the addition of 50 µL 0.4% NADPH (Sigma Aldrich, CAS Nr 2646-71-1) in 0.5% NaHCO_2_ (ChemPur, CAS Nr 144-55-8) and 20 µL 0.15% 5,5′-dithiobis (acid 2-nitrobenzoid) (Sigma Aldrich, CAS Nr 69-78-3). After gentle mixing, 20 mL of glutathione reductase solution with an activity of 6 units per ml was added.

The increase in absorbance at 412 nm was tested for 10 min with a frequency of 1 min. At the same time, a calibration curve for glutathione was made in the range of 0.05 to 10 mmol. The same procedure was repeated, the only difference was that the concentrated 50 mM GSH solution was added to the reaction mixture to obtain the appropriate concentration.

### 2.8. Statistical Analysis

The level of attached glutathione was evaluated using an enzymatic. At least three independent experiments with several examined segments of the samples provided data for the statistical evaluations. The results are presented as a MEAN ± standard deviation (SD). The obtained results were analyzed by one-way ANOVA analysis with a significance level of *p* < 0.05. Statistical analysis was performed using Excel with Office 365 software.

## 3. Results

The properties of the obtained graphene samples were characterized based on the already published research methodology and the results were compared to the previous ones so as to be sure that the tested material was of the same quality [16,17,23]. Figure 1a shows the morphology of graphene after transfer with iron chloride, while Figure 1b shows the local image of graphene after hydrogen delamination.

The Raman spectrum of perfect graphene has three characteristic peaks: G (approx. 1580 cm^−1^), G* (approx. 2450 cm^−1^) and 2D (approx. 2700 cm^−1^).

The G band is doubly degenerate (iTO and LO) phonon mode (*E*_2*g*_ symmetry) at the BZ center that is Raman active for sp^2^ carbon networks. The 2D band is due to an intervalley double resonance (DR) process which involves an electron with wave vector *k* in the vicinity of the Dirac K point and two iTO phonons with wave vectors *q ≈ 2k*. The G* band is also due to an intervalley *q ≈ 2k* process which involves one iTO and one LA phonon [34,35].

Graphene produced in large-scale processes often contains defects that appear in the Raman spectra in the form of additional peaks D (approx. 1350 cm^−1^), D’ (approx. 1620 cm^−1^) and D + G (approx. 2950 cm^−1^). In the case of the D and D’ bands two scattering processes consist of one elastic scattering event by defects of the crystal and one inelastic scattering event by emitting or absorbing a phonon. The difference between the DR process for the D and D’ bands is that, for D band it is inter-valley process because it connects points in circles around inequivalent K and K’ points in the first Brillouin zone of graphene. However, the DR process responsible for D’ band is an intra-valley process, since it connects two points belonging to the same circle around the K point (or the K’ point). The Raman feature at about 2950 cm^−1^ is associated with D + G combination mode and also is induced by disorder [34,36].

The number of defects in graphene have an effect on the shape and nature of the 2D band, and is manifested by decreasing intensity and at the same time increasing its full width at half maximum (FWHM). Figure 2 shows the spectra of unmodified graphene (A) and graphene subjected to the plasma hydrogenation process (B). The spectra were normalized to the G peak. In both spectra, the peaks indicating high concentration of defects are clearly visible (D and D’ peaks present). Plasma modification significantly increased the number of defects in the material, as evidenced by the increase in the intensity of the D and D’ bands, as well as the clear scratching of the D + G band. In the case of modified graphene, a decrease in the intensity of the 2D band and an increase in its FWHM were also observed (Table 1). According to the methodology presented by Concado et al. [37], the distances between defects were calculated for both tested samples. The results show that the plasma hydrogenation process reduced the distance between defects in graphene. Referring to the research conducted by A. Eckmann et al. on the relationship between the ratio of the intensity of the I_D_/I_D’_ peaks and the nature of the defects, it can be concluded that the plasma modification of graphene leads to the concentration of defects similar to grain boundaries [38].

Figure 3 shows a list of several FTIR spectra of unmodified graphene (A), graphene after modification with hydrogen plasma (B), graphene after oxidation in H_2_SO_4_ (C), graphene after modification with hydrogen plasma and after oxidation in H_2_SO_4_ (D) and graphene after modification with hydrogen plasma and H_2_SO_4_ with attached glutathione (E). In spectrum A, the only maximum that appeared is the absorption band of C=C bonds stretching at 1612 cm^−1^, which is characteristic for the system corresponding to the graphene structure [39]. Appearance of the peaks in the range of 3100–3000 cm^−1^ proves that hydrogen is attached directly to the aromatic rings. Absorption bands in the range of 3000–2800 cm^−1^ confirm the presence of aliphatic bonds containing –CHx groups, which were formed on the defects of graphene. Additionally, the presence of these bonds is confirmed by the presence of a band in the range of 1480–1412 cm^−1^. A wide band in the wavenumber range 3700–3400 cm^−1^, belonging to the stretching vibrations of O–H bonds, was present in the spectra of all modified graphene samples. In the case of graphene treated with strong acids, two additional peaks originating from the stretching vibrations of the C=O bonds in two different systems were visible at wavenumbers ca. 1869 and 1718 cm^−1^., The first is similar to the structure that occurs in carbonyl groups, and the second is characteristic for ketones. These two bands were also visible in the spectra C, D, and E.

On the other hand, in the spectrum D, for the sample which was first hydrogenated and then oxidized with strong acids, both bands related to the effect of hydrogen on the sample and new grafted C=O groups were visible. In this case, the intensities of the peaks originating from the C=O groups were much higher as compared to the peaks belonging to these groups in the sample exposed to the acids alone. This confirms the effectiveness of the performed oxidation process. In this case, oxygen adhered to both the random defects in the graphene structure and to those deliberately formed during the hydrogenation process.

Graphene sample after the double modification process was treated with glutathione in order to check the effectiveness of attaching this tripeptide to the surface. Typical peaks for this peptide appeared in the spectrum. Stretching vibrations of N–H bonds originating from the amine group at 3400–3200 cm^−1^ and two peaks at 1561 and 1519 cm^−1^ (deformation vibrations) visible the intensity of the bands originating from C–H vibrations also increased significantly. Moreover, it noticeably widened towards lower wavenumbers, which suggests the presence of additional peaks, including the peak at 2620 cm^−1^, which is related to the presence of the –SH group characteristic for cysteine.

Another very characteristic peak for this peptide is the maximum at 1670 cm^−1^, derived from the amide bond, and at 1405 cm^−1^, belonging to the stretching vibration of the C–N bond [40]. The appearance of such a system of bonds confirms the presence of glutathione on the graphene surface, attached via C=O bonds formed through the applied modification. 

The next step after characterizing the material was to conduct experimental tests in order to optimize the process of attaching glutathione to the surface of graphene samples. As a result of these works, the original procedure developed by L.Mu was modified [32]. Namely, we used glutathione solutions in the range of 5–100 mM and we changed the incubation time in the crosslinking mixture. Finally, based on studies comparing individual chemical surface modifications, it was decided that a 40 mM glutathione solution with a 3 h activation time using the EDAC-NHS mixture would be used.

The key parameter in the study was the characteristics of the relationship between the type of chemical modification of the graphene surface and the level of added glutathione. At the same time, glutathione binding stability was checked depending on the surface and storage conditions of the sample.

Figure 4 shows a comparison of the level of glutathione connected to the graphene obtained by delamination method, which were activated in sulfuric acid in the preparatory stage. As the control sample, the graphene obtained by the delamination method without subsequent chemical modification was used. The presented results clearly show more effective binding of glutathione to the graphene surfaces activated with 75% sulfuric acid as compared to the modification with 50% sulfuric acid. A similar level of glutathione attachment was achieved for graphene surface activated in 50% nitric acid.

An important factor being a subject of the optimization was the binding time of glutathione to graphene through the EDAC-NHS complex, as well as the durability of this functionalization, which translates into the conditions in which the surfaces modified by glutathione attachment should be stored.

In Figure 5 the results for the surface modified in the process of the preparation with 50% H_2_SO_4_ are shown. An abrupt increase in the amount of glutathione attached to the surface of the samples incubated for 48 h is visible. However, very interesting results were obtained during attempts to determine the bond durability, which is particularly important in terms of the potential technological use of the modified graphene surfaces. Samples with attached glutathione were sealed in containers and placed at a temperature of 4 °C. The individual batches of sample were placed in phosphate buffer (10 mM, pH 7.4) or in an air atmosphere (after preliminary drying).

The measured glutathione level after the 24-h incubation showed a different surface behavior of samples stored under different conditions. In the case of dry conditions, a decrease in the detected amount of glutathione was observed for all tested surface types, wherein the observed decrease in the concentration of GSH did not exceed 10% (see Figure 5 and sample S50%). The samples stored in phosphate buffer almost completely lost the attached glutathione (see Figure 5 and samples S50% and S75%). At the same time, the concentration of glutathione in the buffer was determined. As a result, the detected glutathione level was close to that lost by the analyzed surfaces. The same observation was registered for all tested surfaces.

Another interesting phenomena related to the preparation of the surface with the use of strong inorganic acids was determined. The observed increase in the level of glutathione adhered to the surface after 48-h incubation time was at a similar level for all tested surfaces (Figure 6 shows an example comparison between samples N50% and S75%, whereas in Figure 5 it is sample S50%). In the case of extending the incubation time above 48 h, the observed increase in glutathione level did not show statistical significance.

In the case of graphene samples obtained using the delamination technique, a different nature of the correlation between the method of surface preparation and the level of bonded glutathione can be seen. As shown in Figure 7, for this material, no performance limitations in the adhesion process associated with the use of 50% H_2_SO_4_ were observed, and the level of attached glutathione for both tested sulfuric acid concentrations was similar. The obtained data show a similar relationship for two different methods of storing the samples after the GSH attachment. Also, in this case, the stability of glutathione binding to the graphene surface is visible when the sample is stored in dry conditions. A contrary effect can be observed when the samples are left in an aqueous environment (phosphate buffer). Namely, the loss of the attached glutathione is observed. The same relationship was observed for the graphene samples obtained by the delamination method and modified with nitric acid. In this case, the attached glutathione remained stable after 24 and 48 h of storage under dry conditions, while the water conditions caused the loss of glutathione from the surface of the graphene samples.

## 4. Discussion

Graphene is a modern material which, due to its unique properties, has a great application potential, particularly in the field of biomedical engineering. However, the effective use of graphene requires modification of its surface, for example through functionalization by selectively attaching particular molecules with desired biochemical properties or importance for cell physiology [11,41,42].

There are increasing reports in the scientific literature about the modification of graphene by attaching proteins or peptides to its surface, but the majority of this work focuses on showing the diversity of attached bioactive molecules, and teams usually do not have graphene obtained and activated by various methods at their disposal. The presented results allow the analysis of the possibilities of functionalization of the graphene surface, considering the differentiation of the technology of obtaining a “raw” graphene and the preliminary surface preparation before the adhesion reaction of biologically active molecules.

An example of such a molecule is glutathione, a peptide of three amino acids that is of great importance in redox reactions taking place in the cell. It acts as a scavenger of reactive oxygen species and free radicals, and at the same time, with the participation of glutathione transferase, it is an important element of cell protection against xenobiotics. It performs this function thanks to the central amino acid, cysteine, which due to the -SH groups is able to form sulfide bridges [41,43,44]. This makes glutathione both a potential functional molecule that can be used in signaling reactions as well as a very convenient model for optimizing the attachment processes of peptides and proteins to graphene surfaces. Extremely important for the application as a model molecule is the possibility of using highly sensitive enzymatic methods to measure the concentration of glutathione, and thus the measurement of even small differences in the number of molecules adhered to the examined surfaces [33].

The first reports on the adhesion of proteins to graphene surfaces began to appear in literature globally after 2010 [42,45,46], with the following years resulting in an increased number of publications. Most authors have focused on the development of a procedure for attaching biologically active molecules, such as enzymes, in order to create graphene surfaces that can be part of biosensors [42,47,48,49].

At the same time, despite the growing interest in such modifications of graphene surfaces, only single studies have focused on the characteristics of the interactions between the elements of proteins and peptides and the surface of the carbon material, and they focus on the differences in interactions related to the diversity of protein structures [50,51,52]. In our work, a different approach was used. Namely, we used one type of peptide (glutathione) as a starting point, and we focused on the analysis of the effectiveness of the peptide attachment reaction depending on the methodology of preparation of the graphene surface.

The use of techniques typical for material engineering research allowed to characterize the obtained graphene—the use of FTIR techniques and Raman spectroscopy enabled an analysis of both the original carbon material and the changes in its characteristics occurring during glutathione functionalization (Figure 4). The most important element was the increase in the intensity of peaks in the range of 3400–3200 cm^−1^ wavenumbers, typical for the presence of N-H bonds, and the increase in the absorption bands at 1561 and 1519 cm^−1^, confirming the increase in the C-H bond share on the graphene surface. Such results are consistent with the results obtained by other research groups [53,54].

The comparison of graphene surface activation processes through the use of strong inorganic acids shows that it is possible to optimize the technology by using sulfuric acid with a lower concentration (50%). The analysis of the effectiveness of surface biofunctionalization as a result of glutathione adhesion shows that the obtained process efficiency is at a similar level (Figure 7 and Figure 8), and the use of acid with a lower concentration is of great application importance, both from the point of view of technological safety and from the point of view of the costs of the reagents used. However, this effect was achieved for the graphene obtained in the delamination process, while the graphene obtained by chemical etching underwent much lower activation and the peptide attachment process achieved a lower efficiency (Figure 8). Similarly, differences in the efficiency of glutathione attachment were also visible for activation with 75% sulfuric acid, but the observed differences were of a lower magnitude. In the case of activation of the graphene obtained by chemical etching, the optimal activation method seems to be the use of nitric acid (50%). This is an interesting observation, but the mechanism responsible for the presented differences is currently unknown and requires further experimental work.

The conducted works also allowed to analyze the durability of the connection of glutathione with the graphene surface. Glutathione has already been used to functionalize the graphene surface, however, both the work of the team led by N.D. Luong [55] and the work of L. Mu et al. [32] did not cover this type of research, focusing mainly on the development of the methodology of adding thiol derivatives. Particularly with regard to the work published by L. Mu et al. [32], it is important to note that the slightly modified methodology allows for achieving a stable connection with glutathione, with the simultaneous possibility of its disconnection by dissociation after placing the functionalized surface in an aqueous solution. This creates new opportunities to use the developed graphene modification, such as enhancing its antibacterial properties by releasing glutathione responsible for inhibiting free radical reactions that may disturb the redox balance during bacterial infection [41,56].

## 5. Conclusions

The applied technology of graphene surface modification allows for binding of peptide molecules. In addition, this bond is stable during storage in dry conditions, which is extremely important in the context of the use of functionalized graphene surfaces in biosensors.

The surface activation with use of 50% nitric acid allows an increase in the efficiency of glutathione attachment of more than six times in the case of graphene from chemical etching and both sulfuric acid concentrations for delaminated graphene.

The conducted research has shown that the preparation of the surface for glutathione binding with sulfuric acid occurs with similar efficiency with the use of 75% and 50%. This allows the use of acid with a lower concentration, optimizing the process both in terms of cost, technology, safety, and environmental harm.

## Figures and Tables

**Figure 1 nanomaterials-11-00756-f001:**
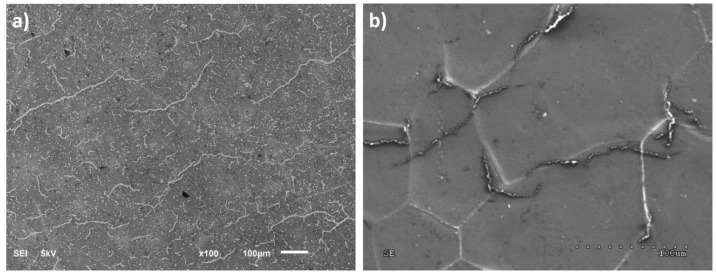
SEM image of graphene: (**a**) transferred to a SiO_2_ substrate after chemical etching, (**b**) on PMMA after hydrogen delamination.

**Figure 2 nanomaterials-11-00756-f002:**
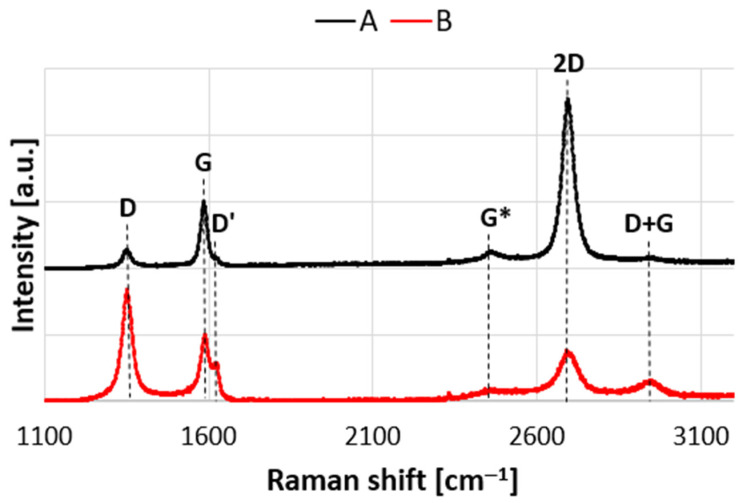
Raman spectra for unmodified graphene (A) and graphene subjected to plasma hydrogenation (B).

**Figure 3 nanomaterials-11-00756-f003:**
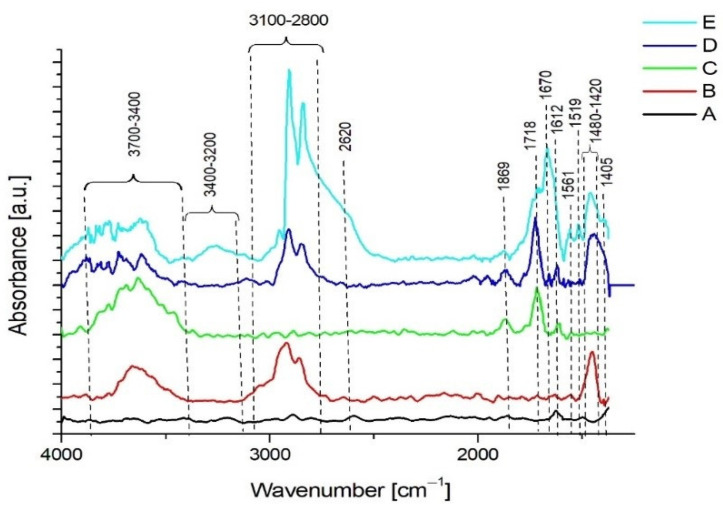
Summary of FTIR spectra for subsequent graphene modifications—unmodified graphene (A), graphene after modification with hydrogen plasma (B), graphene after oxidation in H_2_SO_4_ (C), graphene after modification with hydrogen plasma and after oxidation in H_2_SO_4_ (D), graphene after modification with hydrogen plasma and H_2_SO_4_ and attached glutathione (E).

**Figure 4 nanomaterials-11-00756-f004:**
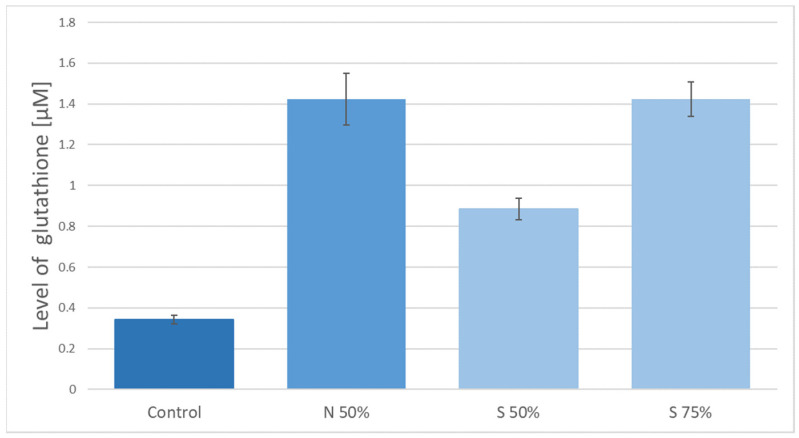
The level of glutathione on the surface of graphene obtained by delamination (N50%—50% nitric acid, S50%—50% sulfuric acid, S75%—75% sulfuric acid).

**Figure 5 nanomaterials-11-00756-f005:**
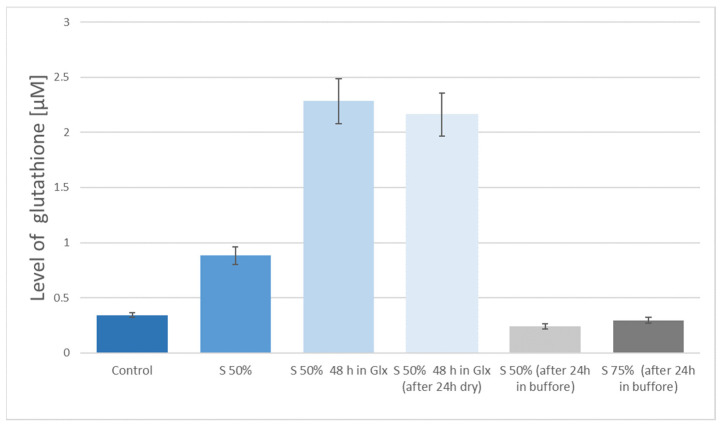
The level of glutathione on the surface of graphene obtained by delamination method and modified by H_2_SO_4_ (S50%—50% sulfuric acid, S75%—75% sulfuric acid, Glx—glutathione solution).

**Figure 6 nanomaterials-11-00756-f006:**
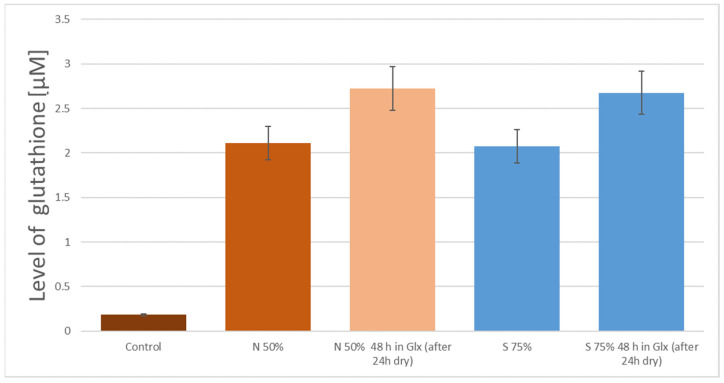
Comparison of adhered amount of glutathione to graphene samples stored in dry conditions (N50%—50% nitric acid, S75%—75% sulfuric acid, Glx—glutathione solution).

**Figure 7 nanomaterials-11-00756-f007:**
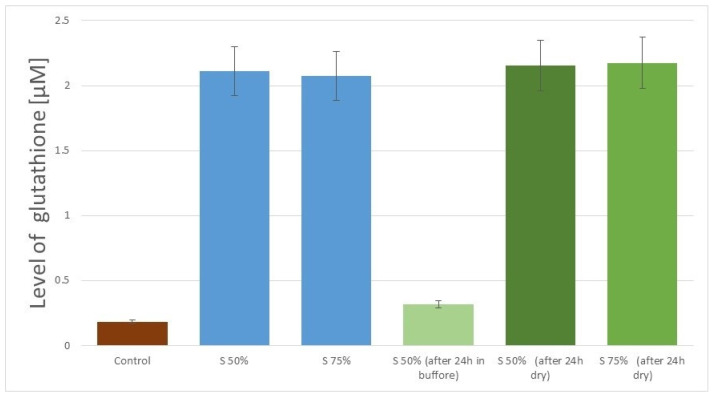
Amount of adhered glutathione to activated graphene surface by sulfuric acid (S50%—50% sulfuric acid, S75%—75% sulfuric acid).

**Figure 8 nanomaterials-11-00756-f008:**
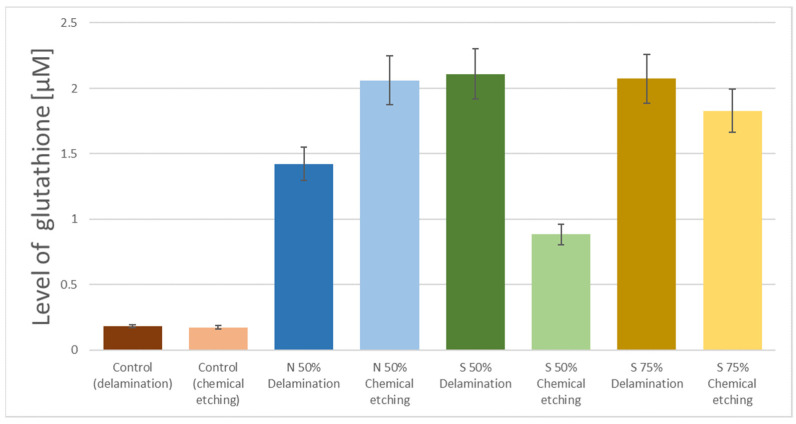
A comparison of activation with inorganic acids for both types of graphene (N50%—50% nitric acid, S50%—50% sulfuric acid, S75%—75% sulfuric acid).

**Table 1 nanomaterials-11-00756-t001:** The ratios of the intensity of the characteristic peaks and the FWHM value calculated from the deconvolution of the Raman spectra.

	Reference Sample	Graphene after Hydrogenation
I_D_/I_D_’	5.43	4.14
I_D_/I_G_	0.26	1.76
L_D_ [nm]	26.46	10.22
I_2D_/I_G_	2.56	0.74
FWHM [cm^−1^]	40.97	75.04

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
