# Peer review of "Optimization of Glutathione Adhesion Process to Modified Graphene Surfaces"

_nanomaterials, 2021, doi:10.3390/nano11030756_

Round 1
Reviewer 1 Report
The manuscript is original by reporting the optimization of the glutathione adhesion process to modified graphene surfaces.
The text should be checked for typos. Ex: pag.4 – “…pH 7, 0 containing…”.
Besides that, a comprehensive characterization is performed with sound scientific method and appropriate discussion.
However, both the abstract and conclusions should be more elaborate and present a more clear picture of the work in a quantifiable fashion.
A graphical abstract would be useful.
Author Response
1a. The text should be checked for typos. Ex: pag.4 – “…pH 7, 0 containing…”.
Many thanks for your comment. We have checked the text to eliminate editing errors. We believe that now the manuscript will be more enjoyable to read.
1b. Besides that, a comprehensive characterization is performed with sound scientific method and appropriate discussion.
However, both the abstract and conclusions should be more elaborate and present a more clear picture of the work in a quantifiable fashion.
A graphical abstract would be useful.
We are aware that a graphical summary could have been helpful, however, due to the voluminous nature of the paper along with the large number of diagrams, we deliberately refrained from using it.
At the same time, following your suggestion, we have modified the conclusions to better reflect the results obtained during the experimental work.
Reviewer 2 Report
The topic is urgent and the results are interesting, nevertheless, the manuscript in its current form is not suitable for publication. The authors have to answer the questions and correct the mistakes.
- What is the basis of the protocol used for chemical activation of graphene (lines 117-122)? The authors do nor refer to any literature sources, so, why such reagents ant their concentrations were chosen? The authors‘ claim that these acids create "bridges" between the graphene and the amino acids should be confirmed by the XPS data.
- The theoretical basis for activation of graphene of using hydrogen should be described in more detail. The authors contrast this process to the chemical activation with acids (line 126), nevertheless, they provide neither an explanation nor evidence of what processes are going on that procedure.
Author Response
The topic is urgent and the results are interesting, nevertheless, the manuscript in its current form is not suitable for publication. The authors have to answer the questions and correct the mistakes.
- What is the basis of the protocol used for chemical activation of graphene (lines 117-122)? The authors do nor refer to any literature sources, so, why such reagents ant their concentrations were chosen? The authors‘ claim that these acids create "bridges" between the graphene and the amino acids should be confirmed by the XPS data.
- The theoretical basis for activation of graphene of using hydrogen should be described in more detail. The authors contrast this process to the chemical activation with acids (line 126), nevertheless, they provide neither an explanation nor evidence of what processes are going on that procedure.
Many thanks for your comments - this will enable us to improve the manuscript, both in terms of content and editing. We have done our best to eliminate all text editing errors, such as incorrect spacing or spelling mistakes.
The method of chemical activation of graphene we used was developed by our own and has been previously published. Our oversight is the lack of a literature reference : Kaczmarek, Ł., Kula, P., Warga, T., Kołodziejczyk, Ł., Louda, P., Borůvková, K., Niedzielski, P., Szymański, W., Voleský, L., Pawłowski, W., & Zawadzki, P. Creation of a 3d structure based on the high strength metallurgical graphene® Surface Review and Letters Vol. 26, No. 06, 1850206 (2019) (https://doi.org/10.1142/S0218625X18502062). The proper reference has been added.
For the studies described in the manuscript, modified acid concentrations were used. This is related to parallel experimental work results which have not yet been published.
In the revised version of the manuscript, the description has been moved from subsection 2.3 to subsection 2.4.
In response to objections concerning the use of the term "bridges" between the graphene and the amino acids", we would like to clarify that these bridges in our case are the newly formed C=O groups described in lines 257-264 of the original version of the manuscript (in the revised version these are lines 261-268).
Further confirmation that 'bridges' provide the basis leading to the attachment of glutathione by the formation of a 'new' amide bond is described in lines 273-277 of the original text (lines 277-281 in the modified version of the manuscript). In our experience with other graphene research projects, the use of the XPS method does not allow for a clear demonstration of this type of bonding.
Further confirmation of this type of bonds obtained during the modification processes was made with use of FTIR analysis. The bands shown in the spectrum corresponding to the particular types of bonds are, in our opinion, an evidence of the effectiveness of this modification.
Reviewer 3 Report
Dear Authors,
Congratulations on your manuscript entitled "Optimization of glutathione adhesion process to modified graphene surfaces". It is well-written. There are a few pointers for your consideration to make it even better.
1) Line 28, consider adding a comma, after decade.
2) Add references for line 38. For references [4-6], consider referencing them individually.
3) Around line 48, maybe can discuss the toxicity of graphene.
4) Remove extra spacing in line 68.
5) Consider quoting chemical assisted exfoliation of graphene oxide document S0254058412007110 in line 69.
6) Check line 95. There is a division symbol?
7) A diagram in section 2.1 production of graphene will add interest to the manuscript.
8) The sentence in line 125 is not clear. Consider revising?
9) In line 177, pH 7,0. Do standardize 7.0 throughout the manuscript.
10) Remove lines 186-188.
11) Good characterization, keep it up!
12) Consider referencing document "c5ra05854f" in line 389.
13) In line 413, check the spelling "delamination".
Author Response
Congratulations on your manuscript entitled "Optimization of glutathione adhesion process to modified graphene surfaces". It is well-written. There are a few pointers for your consideration to make it even better.
1) Line 28, consider adding a comma, after decade.
2) Add references for line 38. For references [4-6], consider referencing them individually.
3) Around line 48, maybe can discuss the toxicity of graphene.
Thank you for your comment, but for the sake of the coherence of the text, We prefer to leave it as it is. In the case of the biocompatibility of graphene, We have relied on the literature data and have supplemented the text with references to the two most relevant.
4) Remove extra spacing in line 68.
5) Consider quoting chemical assisted exfoliation of graphene oxide document S0254058412007110 in line 69.
Thank you for this comment. The proposed literature reference is very interesting, I have read it and added a reference to this article.
6) Check line 95. There is a division symbol?
7) A diagram in section 2.1 production of graphene will add interest to the manuscript.
Details of the HSMG graphene manufacturing process can be found in our previous works and patents [14–17,24,25]
8) The sentence in line 125 is not clear. Consider revising?
The entire description of graphene hydrogenation has been rewritten
9) In line 177, pH 7,0. Do standardize 7.0 throughout the manuscript.
10) Remove lines 186-188.
11) Good characterization, keep it up!
12) Consider referencing document "c5ra05854f" in line 389.
13) In line 413, check the spelling "delamination".
Many thanks for your comments - this will enable Us to improve the manuscript, both in terms of content and editing. We have done our best to eliminate all text editing errors, such as incorrect spacing or spelling mistakes.
Reviewer 4 Report
Article” Optimization of glutathione adhesion process to modified 2 graphene surfaces” deals with functionalization of the graphene surface to bind glutathione. The authors wrote this article thoroughly. The spectroscopic part is don well Authors mentioned the more exact indicator ID/ID’ ratio and not ID/IG. FTIR peaks are also assigned well. I'm afraid I have to disagree with the first sentence of conclusions “The applied technology of graphene surface modification allows for permanent bind-434 ing of peptide molecules.” I would change it to something like we have confirmed binding for at least 48h ……(and not permanent you did not prove permanent)
I found some minor text errors which should be corrected relatively fast by authors. After corrections, I recommend accepting it for publication.
“The proper nomeclature seems to be 67…” Correct nomenclature
“In the next stage of transfer, the foil with graphene was 106 transferred to Si or Si/SiO2 substrate.” Twice transfer/transferred, replace one of them
“For 131 this purpose, after the transfer using thin PMMA foils, the graphene was hydrogenated 132 in a plasma-chemical reactor.” What was the temperature, type of gas, pressure, the voltage applied, etc., of this process?
The discharge electrodes were placed on insulating supports 134 a few centimeters above the surface of the table. I do not understand. I assume that graphene on PMMA was placed on one of the electrodes at lower pressure in a specific gas atmosphere. Afterwhich high voltage was applied. This would give you a low-temperature plasma discharge.
2.5.1. RAMAN Capital letters? 2.5.2. FTIR Full names will help
“…1100–3100 cm-1 wavenumber range. The obtained raman spectra 151” top index
ddH20 Does this water correspond to MiliQ? Add resistivity if you find it. Please also add in Material and methods Companies providing chemicals with CAS numbers.
2.6. Glutathione attachment procedurę should be above 2.5. Surface characteristics
“…concentrated50 183 mM GSH solution…” add space
Figs 1 and 2 do not provide too much information. Even scales are different. If you insist on having them, maybe show it as Fig 1 a b
Figure 5 6 7 8 9 How many samples were used to get these values in column plots? If you did it multiple times, you can perform variance analysis like it is used usually in biology and get the p-value (statistical significance of evidence, possible to do in Origin ANOVA package)
“In the case of ex-336 tending the incubation time above 48 hours, the observed increase in glutathione level did 337 not show statistical significance. 338 “ How were the error bars counted?
Author Response
Article” Optimization of glutathione adhesion process to modified 2 graphene surfaces” deals with functionalization of the graphene surface to bind glutathione. The authors wrote this article thoroughly. The spectroscopic part is don well Authors mentioned the more exact indicator ID/ID’ ratio and not ID/IG. FTIR peaks are also assigned well. I'm afraid I have to disagree with the first sentence of conclusions “The applied technology of graphene surface modification allows for permanent binding of peptide molecules.” I would change it to something like we have confirmed binding for at least 48h ……(and not permanent you did not prove permanent)
Many thanks for your comments - this will enable us to improve the manuscript, both in terms of content and editing. We have done our best to eliminate all text editing errors, such as incorrect spacing or spelling mistakes.
We agree with the comment on 'permanent glutathione binding' and this term has been removed from the manuscript. During the experimental work, samples stored for 7 days were also checked and in their case glutathione was visible on the surface, however very few samples were anlayzed to be sure that glutathione was permanently attached. This observation is very interesting and we will carry out additional tests to determine the durability of such binding and the useful life of the modified graphene surface. Unfortunately, it is not possible to carry out the planned work and incorporate the results into the present manuscript.
“For 131 this purpose, after the transfer using thin PMMA foils, the graphene was hydrogenated 132 in a plasma-chemical reactor.” What was the temperature, type of gas, pressure, the voltage applied, etc., of this process?
The discharge electrodes were placed on insulating supports 134 a few centimeters above the surface of the table. I do not understand. I assume that graphene on PMMA was placed on one of the electrodes at lower pressure in a specific gas atmosphere. Afterwhich high voltage was applied. This would give you a low-temperature plasma discharge.
With reference to the reviewer's remarks, the description of the process of graphene hydrogenation has been rewritten and completed.
Companies supplying chemicals with CAS numbers have been completed and full spectroscopy names introduced
ddH20 Does this water correspond to MiliQ? Add resistivity if you find it. Please also add in Material and methods Companies providing chemicals with CAS numbers.
Companies supplying chemicals with CAS numbers have been completed and full spectroscopy names introduced. The distilled water used to prepare the solutions was of water quality MiliQ.
Figs 1 and 2 do not provide too much information. Even scales are different. If you insist on having them, maybe show it as Fig 1 a b
Thank you for this comment. SEM pictures were combined in one figure a) and b).
Figure 5 6 7 8 9 How many samples were used to get these values in column plots? If you did it multiple times, you can perform variance analysis like it is used usually in biology and get the p-value (statistical significance of evidence, possible to do in Origin ANOVA package)
The information on statistical analysis has been completed - chapter 2.9 added. All trials were performed in three independent replicates and statistically analyzed using ANOVA test.
“In the case of ex-336 tending the incubation time above 48 hours, the observed increase in glutathione level did 337 not show statistical significance. 338 “ How were the error bars counted?
Error bars were calculated using an excel spreadsheet - data from three independent replicates were used to calculate the standard deviation.
Round 2
Reviewer 2 Report
The manuscript can be accepted for publication